# The tangled ways to classify games: A systematic review of how games are classified in psychological research

**Jolanta Starosta[1], Patrycja Kiszka[2], Paulina Daria Szyszka[3], Sylwia Starzec[2], Paweł Strojny[1] ***

**1** Faculty of Management and Social Communication, Institute of Applied Psychology, Jagiellonian University, Kraków, Poland, **2** Doctoral School in the Social Sciences, Jagiellonian University, Kraków, Poland, **3** Faculty of Philosophy, Institute of Psychology, Jagiellonian University, Kraków, Poland

* p.strojny@uj.edu.pl

**Data Availability Statement:** All relevant data are within the manuscript and its Supporting Information files.

## Abstract

In the face of the rapid evolution of the gaming market and the puzzling overlap of genres, consistency in classification seems elusive. The purpose of the present review was to explore the classification of video game genres in the context of psychological research. The aim was to address the challenges associated with creating consistent and meaningful classifications of video game genres, considering the rapid evolution of the gaming market and recent tendency to create games that could be classified into multiple genres. We performed a search in four databases according to the PRISMA guidelines and reviewed 96 full-text papers (N = 49 909). Through our findings, we reveal how researchers strive to classify genres and the numerous complications that arise from this pursuit. In the face of these challenges, we propose alternative ways of classifying genres. Our first proposal is a new classification of video game genres based on our literature review. In our second proposal, we advocate a more detailed understanding by focusing on specific gaming mechanics, and thus we introduce the innovative concept of utilizing community-based tags, such as Steam tags, as an alternative to genres in psychological research.

## 1. Introduction

Over the last two decades of technological development, video games have become one of the most popular ways to spend free time. In 2023, there were nearly 3.4 billion people playing video games, and these numbers will only increase in the future [1]. Over the years, the researchers have been investigating the associations between video game genres and other aspects of human functioning such as gaming disorder (GD), motivational factors, and cognitive functions [2–4], to name only three examples. However, the complexity of the fast development of the game industry has led to difficulties in creating separable genre categories and has tangled the ways of classifying modern games. This lack of consistency in categorizing games into specific genres could be present in scientific research as well, in which the same games can be included in totally different categories, making the process and results of classification less transparent. The purpose of this paper is to explore recent difficulties in using video

**Funding:** This research is a part of SONATA Bis-11 research project "The transformation process from gaming involvement to gaming disorder: Delineating social and motivational antecedents from consequences", no. 2021/42/E/HS6/00068, funded by The National Science Centre (NCN) and carried out at the Faculty of Management and Social Communication, Jagiellonian University, Kraków, Poland. The publication of this study in the Open Access format was financed by the Strategic Programme Excellence Initiative at the Jagiellonian University in Kraków.

**Competing interests:** The authors have declared that no competing interests exist.

game' genre classification in scientific work and provide a proposal for a coherent systematics of games, allowing for further exploration of significant differences between individual titles in other aspects of functioning.

## 1.1 Difficulties in defining genres

Undoubtedly, the gaming industry has dramatically changed over the years, especially when it comes to game genres and mechanics [5]. In 1980, due to the economic pressure in the gaming industry, games started to be categorized into sets of genres that shared narrative structure, gameplay mechanics, dynamics, and schemes. One of the first video game genre classifications was created by Crawford [6]. "Skill-and-action games"—combat, racing, and maze games—were distinguished from "Strategy games", which included roleplaying (RPG), adventure, and educational games. Early taxonomies, even if extensive and elaborated as in Wolf's 42 video game categories [7] or in the typology created by Aarseth et al., [8], could not fit into modern game genres such as Massively Multiplayer Online Role-Playing Game (MMORPG) or Multiplayer Online Battle Arena (MOBA). Another problem was that some taxonomies created by scientists did not fulfill their communicational function and due to that they were not compatible with the market and community [9].

Taxonomical obstacles emerged because of the progress of blurring the lines between game genres. Today's games are harder to distinguish purely by their characteristic mechanics; for example, Action games mechanics can be found in multiple other genres [5]. Such a state led to the creation of multigenre hybrids such as Action-RPG or Action-adventure. After all, the enormously popular *League of Legends* combines elements of real-time strategy and action, which creates the MOBA genre. Another problem emerges from the development of games that combine not two but three or more different genres, such as *Fortnite* or *Minecraft*, which are even more difficult to classify [5,10]. Heintz et al. [11] emphasize that the relation between the genres started to overlap and that the genres themselves are not consistently defined. Even the sellers categorize video games into multiple genres to increase the possibility of someone buying their product. Furthermore, games can be categorized differently depending on which attributes spark the player's interest. For example, *Genshin Impact* can be played for narrative reasons or for the gameplay focused on grinding skills and equipment of the character. Consequently, any given game can be included in one genre on the basis of its gameplay and in another because of its iconography or story [12]. Heintz et al. [11] tried to map the attributes characteristic of extracted game genres–e.g., Adventure includes collecting objects, communicating, emotional connection to character, puzzles, searching, fantasy world, exploration, and story. However, again, it is visible that some of the attributes were shared across multiple genres or even overlapped with each other. Another trend in the gaming industry is creating game genres on the basis of mechanics like "First person" or "Real time strategy" (RTS). The first example was usually connected to Shooters or Action games [4]. On the other hand, RTS and TBS (Turn-based strategy) emerged as new genres from Strategy [13]. Nevertheless, despite these difficulties, psychologists have been trying to classify game genres for decades. This is because game characteristics have repeatedly been shown to be related to psychological phenomena. There is no indication that it will be different in the future; still, understanding the interaction between gamer characteristics, situational conditions, and game features will be the key to understanding and predicting human behavior, thoughts, and emotions.

## 1.2 Game characteristics and their classification are important subjects of psychological research

There are many areas of psychological research where games' features have proven to play a role. Among others, these are: aggressive behavior [e.g., 14], education [e.g., 15], health [e.g.,

16], sexual behavior [e.g., 17], and many others. Two fields seem to be particularly interesting for researchers in recent years: the relationships between games' characteristics and gaming disorder and cognitive development. Below, we briefly present examples of research in these areas to illustrate that it is impossible to ignore the features of games in research on human interaction with games.

**1.2.1 The dark side–gaming disorder.** Scientific exploration of game genres is an important issue since it can provide insight into the specific mechanisms and features of video games that may be associated with the development and maintenance of problematic gaming behaviors. Moreover, a number of studies [e.g., 18,19] have shown that different gaming motivations lead players to reach for certain genres.

According to a systematic review published in 2021 [2], MMORPGs are most frequently linked to symptoms of Gaming Disorder (GD). In the studies subjected to the present review, the authors' proposed definitions of MMORPGs (if any) ranged from limited and less specific, e.g., "online fantasy worlds where players interact with other players" [20], to more elaborate and precise, e.g., "MMORPGs take place in persistent virtual worlds continuing to exist independently of the player's presence. Gamer's avatar has to constantly progress (e.g., to gain levels and items) through in-game achievements, which are generally favored by successful collaborations and/or competitions with other players." [21]. Moreover, none of these definitions are precise enough to exclude non-MMORPGs; for example, *Warhammer 40,000*: *Darktide* or *Minecraft* meet all the criteria but are still not considered MMORPGs.

In describing the types of mechanics that evoke problematic usage in video games, including MMORPGs, Klemm and Pieters [22] list the following: (1) rewards, which Yee [23] characterized as "random-ratio reinforcement schedule based on operant conditioning. Early achievements are quick, almost instantaneous, and gradually take more and more time and effort until progression becomes almost imperceptible" and (2) social integration that includes not only casually talking to other gamers via chat but also forcing them to create groups in order to complete quests that cannot be achieved otherwise, leading to a greater expenditure of time for the purpose of keeping up with others or outpacing them. It is also worth mentioning here that while this combination of features could be considered distinctive for MMORPGs, it is easy to identify non-MMORPGs that use both mechanics together, such as *World of Tanks*.

The specification of mechanics evoking problematic gaming also corresponds to MOBA games. The previously mentioned literature review [2] emphasizes the role of MOBA games, stating that the elevated risk of gaming disorder has been increasingly recorded recently among MOBA gamers [24–27], and additionally reports First-Person Shooters (FPS) and Real-Time Strategy games (RTS) as associated with higher endorsement of GD symptoms alongside MMORPGs and MOBA.

It is worth mentioning that in some of the empirical studies there is a noticeable lack of differentiation between Real-Time Strategy (RTS), which allows players to play simultaneously and has been reported to be associated with GD risk [e.g., 28], and Turn-Based Strategy (TBS), where players take turns to play and which association to GD has been found to be much weaker [28], thus substantially differing in their mechanics and connections to problematic gaming behaviors. One of the definitions of Strategy games (both RTS and TBS as one genre) subject to the present review reads: "[Strategy games] center around strategic decisions to achieve a desired outcome" [29], thus leaving many unknowns. There were more instances of studies not differentiating two kinds of strategy game genres and additionally lacking definitions [30–32].

**1.2.2 The bright side–cognitive abilities.** It is well known that playing video games does not only lead to negative consequences. The literature review [33] on the topic of commercial video games and cognitive functions shows that gamers perform better than non-gamers on

tests of various cognitive functions, such as attention, working memory, visuo-spatial function, probabilistic learning, problem-solving skills, and second language. Additionally, certain game genres are associated with improvement of particular cognitive abilities.

According to the review, playing Action video games is linked to increased visual attention [34–38], better working memory capacity [39], more precise and detailed visual representation [40], efficiency to use visually and auditorily available information [41] higher activation in brain regions involved in visual imagery, semantic memory, and cognitive control [42]. However, in this review, Action video games genre was based on another study [43] and consisted of two sub-genres: (1) First-Person Shooters (FPS) and (2) Third-Person Games (TPG) defined as (1) "VGs [video games] played in the perspective of players" and (2) "VGs [video games] played by using the avatars of the players whose experience relies on the arrangement of space and time in game environments" respectively. This concept of the action game genre is broad and not specific, since as many as three differentiated genres proposed in other studies [19,44,45] could be included in it: Action games, First-Person Shooters and Platformers. The importance of accurate game genres classification has been shown in a study by Dobrowolski et al. [46], where the authors argue that video game genres play a significant role in the cognitive benefits of gaming, particularly highlighting a lack of consistency in the classification of action games.

However, there is a risk that by attributing beneficial effects to Action games, researchers fall into the trap of rigid attachment to classic genres. The claim that "action games are associated with better cognition" can only be half true. From the perspective of previously mentioned cognitive functions development, there are no theoretical reasons to exclude RTS (but not TBS) or MOBA games from the group of potentially beneficial genres. They share with classic Action games features such as dynamic gameplay, a large number of distractors, or the need to make quick decisions. On the other hand, there is no reason to claim that non-MMORPG games that share key characteristics with MMORPGs (random-ratio reinforcement, social relationships, interdependence on other gamers) should be more harmless. That is why the authors of the present review believed that it might be beneficial to the quality of future research to take a close look at how researchers define individual characteristics of games (and genres).

## 2. Method

### 2.1 Search strategy

This systematic review was conducted according to the Preferred Reporting Items for Systematic Review and Meta-Analyses [PRISMA; 47]. The aim of the study was to capture the most recent and the most prominent methods of classifying video game genres in empirical studies in disciplines such as psychology and computer sciences. The search was conducted on the following databases: ScienceDirect, Taylor and Francis, Scopus, and PubMed. The last search was performed on December 11th, 2023. The results were limited to the filters of subject area: Psychology, Psychiatry, Computer Science and Neuroscience. The search strategy combined key terms such as "video game genre" OR "video game classification" OR "video game categories" OR "videogame genre" OR "videogame classification" OR "videogame categories" OR "digital game genre" OR "digital game classification" OR "digital game categories".The databases were searched twice systematically and independently by two reviewers. In case of discrepancies, they were resolved through clarification and consensus. Both reviewers collected data from databases that they had screened. Articles were included in the study if they met the following criteria:

1. The articles were published between January 2012 and December 2023 (inclusively). It was due to the enormous changes in the video games market and the need to present actual video game genre classifications;

2. The articles were peer-reviewed and published in English or Polish;

3. The articles had an empirical nature;

4. The articles were related to the science disciplines such as psychology, psychiatry, neuroscience or computer science;

5. The articles included comprehensive video game classification and thus did focus on the comparison more thantwo genres.

The exclusion criteria were:

1. The studies had theoretical nature;

2. The studies focused on one or two genres;

3. The articles includes genres named specifically for the sake of the study (e.g., violent game genres).

Initially, 1229 results were retrieved, including duplicate results. The titles and abstracts of the results from all the searched databases were screened for relevance, which led to the identification of 176 potentially eligible articles, of which 161 were retrieved. After reading entire papers, 64 articles were excluded from the study due to not meeting the criteria of inclusion. Consequently, 93 full-text articles were included in this review. In addition, on the basis of the analysis of references of the identified articles, 4 additional papers were assessed for eligibility, of which 3 were included in the analysis. As a result, 96 papers were analyzed. Fig 1 provides a flowchart outlining the search for the studies.

## 2.2 Data coding/retrieval

The following data were retrieved from all the eligible articles: list of genres, information whether the authors proposed any subgenres (yes/no), information whether the authors provided definitions of genres (yes/no), and information whether the authors provided sample games belonging to genres (yes/no). Furthermore, sources of classification included in the articles were collected to assess their study-uniqueness. We planned to code the source of classification as follows: "Article reference" (when the authors used a taxonomy previously used in another published work), "Website" (when the authors used a taxonomy published on a specialized website), "Game platform" (when the authors used a taxonomy published on a game selling platform), "Arbitrary: authors" (when the taxonomy was made by the authors themselves for this study, without reference to external sources), "Arbitrary: participants" (when the taxonomy was made on a basis of participants responses, without reference to external sources), "Expert judges" (when experts external to the authors provided the taxonomy), "Arbitrary: not specified" (when the taxonomy source was not specified), "Combination" (when the taxonomy used resulted from the compilation of sources of various nature). One additional category ("Market analysis") emerged during classification when the authors decided to use the market analysis as a basis for the games' taxonomy. Moreover, information about the country and the number of participants by gender who took part in the studies was collected. In some cases, the subject of empirical analysis was data collected without the direct participation of the respondents, in such a situation the authors did not provide the number of respondents, as a consequence, we also did not provide N, we marked such studies as 'not

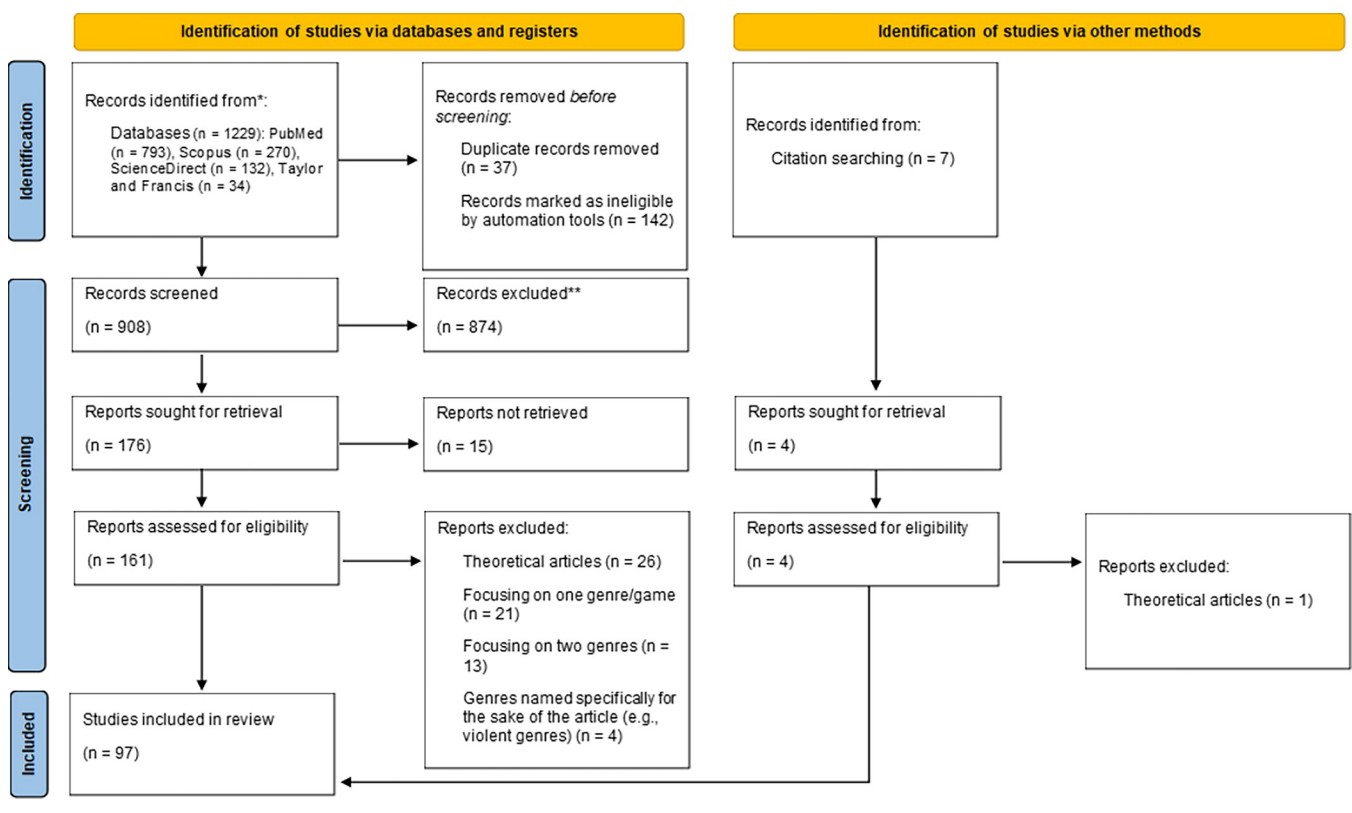

**Fig 1. PRISMA diagram of the search and selection process for the relevant literature.**

applicable'. Impact factor of the scientific journal where articles were published was retrieved to assess the possible quality of the work. All these data are presented in Table 1.

## 3. Results

The studies included in this systematic review were conducted between 2012 and 2023. Over a decade of technological development undoubtedly had an impact on the classification of game genres. The variety of game categories that gained enormous popularity during the last few years mentioned in these studies did not exist before 2012. Some of them might be completely redefined and merge into one big category. Most of the studies were conducted in Europe (n = 40). Twenty-seven articles describe research from North America [20,44,45,71,72,75–77,85,87,91,93,95–97,99,100,103,105,106,108,111,120,123,126–128]. Twelve studies were carried out in Asia [25,33,48,53,54,56,59,64,78,81,113,118]. Eight articles included in this systematic research came from Australia and Oceania [29,52,74,79,80,98,104,119]. Six articles had an international character [31,70,111,121,129,130]. Three studies did not provide information about the origin of the research group [61,69,73]. A total of 89 575 participants took part in the studies included in this systematic research (including 27,88% women). The subgenres were mentioned in sixteen studies [19,48,49,52,53,57,68,76,79,92,99,106,109,112,125,126]. However, four articles included subgenres only for specific genres, e.g., subgenres were retrieved only for RPG, Strategy Games or Adventure [19,48,84,92,112]. Furthermore, two studies provided only one main genre: Action [106] or Social Network Games [126] which were divided into subgenres that were classified as individual genres in other studies, e.g., Sport or Adventure were understood as one of the subgenres of the Action games.

**Table 1. Description of the studies covered in the systematic review.**

| Authors | Impact factor | Country | Participants (% of women) | Genres | Subgenres | Definition | Examples | Source of taxonomy |
|---|---|---|---|---|---|---|---|---|
| Abdullah et al. (2015) [48] | NP | Singapore | 42 (16,67%) | Action-adventure, Adventure, Strategy, Puzzle, RPG, Sport, Simulation | Only for Action-adventure | N | Y | Arbitrary: not specified |
| Alonso-Diaz et al. (2019) [49] | 11.18 | Spain | 404 (42,6%) | Action-adventure (platform, action, graphic adventure game), Strategy and Cognitive (RTS, Puzzle, Mazes and Educational board games), Sports and racing (Sports, Racing, Bat and ball, Shooter), Role and stimulation (RPG, MMORPG) | Y | N | N | Article reference |
| Amiriparian et al. (2019) [50] | 7.25 | Germany | NA | Action or Shooter, Arcade or Platform, Fighting, Racing, Sports, Simulation or World Building | N | N | Y | Game Platform |
| Appel (2012) [51] | 11.18 | Austria | 200 (58%) | FPS, Palor Games, Arcade/jump 'n' run, Action/adventure, Racing, Facebook-games, Simulation, Fantasy/RPG, Sports, Strategy, Activity Games | N | Y | Y | Arbitrary: Authors |
| Azizi et al. (2018) [52] | 1.97 | Australia | 56 (37,5%) | RPG, RTS, Action, Puzzle | Y | N | Y | Article reference |
| Balakrishnan and Griffiths (2019) [53] | 8.0 | India | NA, 25 200 game reviews | Action, Adventure, Arcade, Board, Card, Casino, Casual, Puzzle, Racing, RPG, Sports, Strategy, Trivia, Word | Y | Y | N | Website |
| Bilginer et al. (2021) [54] | 2.072 | Turkey | 320 (58,125%) | Action/Adventure, Sports, Roleplaying, Strategy, Simulation, Shooter, Racing, Fighting, Arcade, Quiz/Trivia, And Classic Board Games | N | N | N | Article reference |
| Boric and Strauss (2021) [55] | NP–CA | Austria | NA | Action, Adventure, Arcade, Fighting, Health & Fitness, Music, Party, Platformer, Puzzle, Racing, RPG, Shooter, Simulation, Sports, Strategy | N | N | N | Combination: Website and Game platform |
| Bowman and Chang (2023) [56] | 2.072 | Taiwan | 875 (41,83%) | PC-Based RPG (Role-Playing Games), Massively Multiplayer Online Role-Playing Games (MMORPG), FPS (First Person Shooters), Action And Adventure Games, Fighting Games, Puzzle Or Leisure Games, Racing Games, Sports Games, Music, Casino Or Gambling Games, Virtual Reality (VR) Games, Augmented Reality (AR) Games, Massive Online Battle Arenas (MOBA), Card Games, Simulation Games, And Open World Games | N | N | N | Arbitrary: authors |
| Braun et al. (2016) [57] | 9.79 | Germany | 2891 (16,25%) | Action, RPG, simulation, strategy | Y | Y | N | Article reference |

*(Continued)*

**Table 1.** (Continued)

| Authors | Impact factor | Country | Participants (% of women) | Genres | Subgenres | Definition | Examples | Source of taxonomy |
|---|---|---|---|---|---|---|---|---|
| Casale et al. (2022) [58] | 1.903 | Italy | 715 (28,5%) | MMORPG, MOBA, FPS, MMO RTS, MMO SPORT, Other | N | N | N | Arbitrary: not specified |
| Cho et al. (2017) [59] | 9.79 | South Korea | 1192 (27,68%) | MMORPG, FPS, Racing, Sports, RTS, Action, Smartphone Game | N | N | N | Website |
| Choi et al. (2020) [33] | 7.08 | China | 110 (63.6%) | Action, Strategy, Miscellaneous (Board/card), RPG, Simulation, Puzzle, Sports, Action-adventure, Adventure, Rhythm | N | N | N | Website |
| Collins and Cox (2014) [60] | 4.87 | England | 491 (48,9%) | Action, Adventure, Casual, Fighting, FPS, MMORPG, Social/Party, Sports, Strategy/Simulation RPG | N | N | N | Arbitrary: not specified |
| Deleuze et al. (2017) [21] | 9.79 | Belgium | 86(0%) | MMORPG, MOBA, FPS | N | Y | Y | Arbitrary: authors |
| Denisova et al. (2019) [61] | 4.87 | NA | 394 (13,2%) | Action RPG, Action-adventure, adventure, Beat 'em' up, Eroge, Fighting, Grand Strategy, Management Simulation, MMORPG, MOBA, Platformer, Puzzle, Roguelike, RPG, Shooter, Sports, Stealth, Survival, Survival horror, Tactical RPG, RBS, Vehicle Simulation | N | N | N | Arbitrary: not specified |
| Dickmeis and Roe (2019) [62] | 1.6 | Belgium | 1170 (53,4%) | Online shooters, MMORPGs, Sports, Strategy, Racing games, Fighting, RPGs, Platform, Puzzle, Adventure, Simulation, Offline shooters | N | N | N | Arbitrary: not specified |
| Dieris-Hirche et al. (2020) [63] | 9.79 | Germany | 820 (26.5%) | MMORPG, FPS, Strategy/simulation, Jump and run, Sport, Beat them u, Adventure, Other | N | N | N | Arbitrary: not specified |
| Dindar (2018) [64] | 11.18 | Turkey | 479 (44%) | Brain and skill, Sports, racing and simulation, Action and adventure, Shooter, Strategy, RPG | N | N | N | Arbitrary: authors |
| Dobrowolski et al. (2015) [46] | 9.79 | Poland | 90 (7%) | FPS, RTS, Platform, Fighting, Adventure, Turn-based strategy, RPG, Racing, Puzzle, MOBA | N | Only FPS & RTS | Only FPS | Arbitrary: not specified |
| Donati et al. (2015) [65] | 6.135 | Italy | 701 (0%) | Management, Browser, RTS, RPG, Action, MMO, Fighting, Arcade/Platform, Retro, Indie, Sports, Casual | N | N | Y | Combination: Article references, Game platforms and Website |

*(Continued)*

**Table 1.** (Continued)

| Authors | Impact factor | Country | Participants (% of women) | Genres | Subgenres | Definition | Examples | Source of taxonomy |
|---------|---------------|---------|---------------------------|--------|-----------|------------|----------|--------------------|
| Elliott et al. (2012a) [44] | 6.135 | USA | 3380 (41%) | MMORPG, Other RPG, Action-adventure, FPS, Other shooter, Sports general, sports other, Rhythm, Driving, Platformer, Puzzle, RTS, Other strategy, Board and card games, Gambling, Other | N | Y | Y | Website |
| Elliott et al. (2012b) [66] | 11.555 | England/ Spain | 3380 (42%) | Action-adventure, MMORPG, Other RPG, FPS, Other Shooter, Gambling, RTS, Other strategy, Board/ Card, Sports General, Sports, Puzzle, Rhythm, Driving, Platformer, Other | N | Y | N | Website |
| Entwistle et al. (2020) [29] | 9.79 | Australia | 1958 (12,1%) | Shooter, Strategy, RPG, Action/Adventure, Simulation, Puzzle, Action, Driving/Racing, Fighting, Sports, GPS (Global Positioning System) | N | N | Only FPS | Article reference |
| Floros and Siomos (2012) [67] | 6.135 | Greece | 2017 (48.3%) | Combat Sim, Sports, Driving Sim, Adventure, MMORPG, Strategy, Martial Arts, Flight Sim, RPG, Mobile/ Handhelds, Platform, Music Sim, Facebook, Life management, Thought | N | N | Y | Arbitrary: not specified |
| Fuster et al. (2016) [68] | 8.957 | Spain | 1074 (5,4%) | Action, Strategy, Role-playing, Adventure, Puzzle, Sports, MOBA, MMORPG | Y | N | N | Arbitrary: not specified |
| Gabbiadini and Greitemeyer (2017) [69] | 3.95 | NP | 392 (56,37%) | Logic, Puzzle, Brain-training, Violent, Platforms, Car racing, Strategy | N | N | N | Arbitrary: not specified |
| Gackenach et al. (2016) [70] | 2.5 | International | 906 (71.9%) | FPS, MMORPG, RPG, RTS, Strategy, Simulation, Adventure, Fightinf, Driving, Sports, Puzzle, Card, Board, Music/dance, Casual | N | N | N | Arbitrary: authors |
| Gilbert et al. (2018) [71] | 4.29 | USA | 273 (0%) | Sport, Action, Online Games | N | Y | Y | Market analysis |
| Green et al. (2017) [72] | 10.172 | USA | 824 (NP) | Action FPS, Action TPS, Action RPG, Action-Adventure, Sports, Driving, RTS, MOBA, Non-action turn based RPG, Fantasy, TBS, Life simulation, Puzzle, Fighting, Phone, Browser, Other | N | N | Y | Arbitrary: authors |
| Guzsvinecz and Szűcs (2023) [73] | 8.957 | NA | NA | Strategy, Puzzle, Adventure, Simulation, Action, Casual, RPG, Sports, Racing, Tabletop, Experimental | N | N | N | Game platform |
| Han et al. (2020) [25] | 5.919 | South Korea | 1532 (32.4%) | RPG, MOBA, Shooting, Simulation, Arcade, Sports, Action | N | N | Y | Arbitrary: authors |

(*Continued*)

**Table 1.** (*Continued*)

| Authors | Impact factor | Country | Participants (% of women) | Genres | Subgenres | Definition | Examples | Source of taxonomy |
|---|---|---|---|---|---|---|---|---|
| Hazel et al. (2022) [74] | 1.837 | New Zealand | 2107 (26.4%) | Fighting, FPS, Hidden Object, MMORPG, MOBA, Mobile/Causal, Music, Point and Click, Puzzle-Action, Puzzle-Other, RPG, Simulation, Strategy, Survival Horror, Text Adventure, TPS, Vehicle | N | Y | Y | Arbitrary: not specified |
| Homer et al. (2012) [20] | 8.957 | USA | 213 (47%) | FPS, Fighting, Sports, Virtual Life World, MMORPG, Puzzle, Party, RTS, Simulation, TBS, Browser, Simple/2D | N | Y | Y | Arbitrary: authors |
| Howe and Cionea (2021) [75] | 2.6 | USA | 435 (56,8%) | FPS, MMO, RPG, Strategy, Social, Sports, Retro | N | N | N | Arbitrary: authors |
| Jang and Byon (2020) [76] | 9.79 | USA | 978 (49.9%) | Imagination, Physical enactment, Sport Simulation | Y | Y | Y | Arbitrary: not specified |
| Jiang and Zheng (2023) [77] | 3,6 | USA | NA, 50 000 games | Adventure, Arcade, Fighting, Indie, Music, Pinball, Platform, Puzzle, Quiz/Trivia, Racing, RPG, Shooter, Simulator, Sport Strategy | N | N | N | Website |
| Jiwal et al. (2020) [78] | 4.02 | India | 233 (36,1%) | ERG, MTG, Others (Adventure, RTS, RPG, MOBA) | N | Y | Y | Arbitrary: authors |
| Johnson et al (2012) [79] | NP—CA | Australia | 466 (18%) | Shooting, Sport/Racing/ Fighting, Action-adventure, Strategy and RPG | Y | N | N | Combination: authors and participants |
| Johnson et al. (2016) [80] | 8.957 | Australia | 543 (18%) | Action-adventure/platformer, Action RPG, MMORPG, RTS and TBS, Shooters, MOBA, Puzzle/Simulation/ Construction, Sport/racing/ fighting | N | N | N | Arbitrary: participants |
| Kim et al. (2022) [81] | 3.752 | South Korea | 3217 (49,15%), 987 games | RPG, FPS, RTS, Racing, Sports, Arcade/Shooting | N | Y | Y | Combination: Article reference and Expert judges |
| Király et al. (2022) [19] | 7.772 | Hungary | 14740 (10,7%) | Shooters, Battle royale, MOBA, Auto chess/Auto battle arena, open world action-adventure, RPG, Online RPG, Strategy, Card, Sport, Simulation, Other | Only for shooters and strategy | N | Y | Arbitrary: authors |
| Kühn and Gallinat (2013) [82] | 13.437 | Germany | 62 (0%) | Building games, Simulation games, Racing games, Ball games, Online roleplay games, Single-player roleplay games, Action-based roleplay games, Click and point adventure games, Action adventures games, Platform games, Ego shooter games, Third person Shooter games, Logik/puzzle games, Arcade games | N | N | Y | Arbitrary: not specified |

(*Continued*)

**Table 1.** (Continued)

| Authors | Impact factor | Country | Participants (% of women) | Genres | Subgenres | Definition | Examples | Source of taxonomy |
|---|---|---|---|---|---|---|---|---|
| Laffan et al. (2016) [83] | 9.9 | International | 207 (43,9%) | RPG, MMORP, Shoot 'em' up, Fighting, Puuzle, Sport, Educational, Action/adventure, MOBA, Sandbox | N | N | N | Arbitrary: participants |
| Lange et al. (2021) [84] | 4.23 | Germany | 484 (42%) | Adventure (action, action-adventure), Beat 'em' up, Casual, Educational, Erotic, FPS, TPS, Music and dance, Opne-world, Platformers, Puzzle, Quiz, RPG, Shoot 'em' up, Simulation, Sport, Strategy, Western | Only for Adventure | N | Y | Website |
| Lee et al. (2014) [85] | NP–CA | USA | NA | RPG, Action/Adventure, Shooter, Strategy, Simulation, Puzzle, Action, Driving/Racing, Fighting, Sports | N | Y | Y | Combination: Website and Article references |
| Lemmens and Hendriks (2016) [86] | 6.135 | Netherlands | 2442 (50,4%) | Action/adventure, Sports, RPG, Strategy, Simulation, Puzzle, Shooter, Racing, Fighting | N | N | N | Combination: Website and Article references |
| Li et al. (2019) [87] | 4.59 | USA | 618 (NA) | MOBA, Shooting/shooters, Battle Royale, Sports-themed, RPG< MMO, Action, Card, Simulation, Strategy | N | N | Y | Arbitrary: authors |
| Li (2020) [88] | NP–CA | Finland | NA | Arcade shooter, Fighting, Soccer, Sandbox, Turn-based Tactic, Resource Management, Music, Shooter, Rogue, Character Action, Strategy, Classic, Board/Card, Gal, Platformer, Survival, MMO, racing, RPG, Strategy RPG, ARPG, Top-down Shooter, Interactive Fiction, Tower Defence, RTS, Simulation, Exploration, Parkour, Education | N | N | N | Combination: Game Platform and Article reference |
| Li and Zhang (2020) [89] | NP–CA | Finland | NA | Strategy/Simulation, Puzzle and Arcade, RPG, Shooter | N | Y | N | Game Platform |
| Lloyd et al. (2019) [90] | 4.65 | England | 252 (39,7%) | Adventure, Fighting, FPS, Music, RPG, Social and Casual, Sports, Strategy, Quiz | N | N | N | Article reference |
| Lonergan and Weber (2019) [91] | 3.26 | USA | 112 (40,2%) | Strategy, Puzzle, Fantasy/RPG, Action-adventure, Sports, Simulation, Racing/speed, Shooter, Fighter, Arcade, Card/dice, Quiz/Trivia, Classic board games, MOBA, Other | N | N | N | Article reference |
| López-Fernández et al. (2021) [92] | 4.102 | Spain | 776 (50,6%) | Action Shooter, Sports, Strategy, Puzzle Brain, Puzzle Skills, Skill Platform, Adventure, Social Sim, Construction, RPG, Fighting | Only for Adventure and RPG | N | Y | Article reference |

(*Continued*)

**Table 1.** (Continued)

| Authors | Impact factor | Country | Participants (% of women) | Genres | Subgenres | Definition | Examples | Source of taxonomy |
|---------|------|---------|------|--------|-----------|-----------|----------|--------------------|
| Mandryk and Birk (2017) [93] | 7.08 | Canada | 491 (44%) | Action, Adventure, Beat them up, Casual, FPS, MMORPG, MOBA, Music, Platform, Puzzle, RPG, Simulation, Sports, Strategy, Vehicle Sim | N | N | N | Arbitrary: authors |
| Manero et al. (2016) [94] | 9.79 | Spain | 754 (45.36%) | FPS, Adventure and thrillers, Singing, dancing or playing instruments, Fighting, Invention or cognitive, Strategy, Sports, racing or simulation, Social and casual, Internet collaborative | N | Y | Y | Combination: Article reference, Website and Data reports |
| Männikkö et al. (2018) [27] | 3.103 | Finland | 465 (37,2%) | Casual, Solo, Vehicle Sim, Strategy and Management, Sport, Shooting Games Online/FPS, MOBA, MMORPG | N | N | Y | Arbitrary: authors |
| Mathews et al. (2019) [95] | 3.912 | USA | 2801 (6,7%) | MMORPG, Online FPS, Offline FPS, Other RPG, Competitive Online Fighting Gams, RTS, Other/TBS, Driving, Sports, Action-adventure/Co-op Action, Board/Card, Survival Horror/Platformer, Puzzle Games/gambling Games with no Currency Betting | N | Y | N | Expert judges |
| Mazurek and Engelhardt (2013) [96] | 2.5 | USA | 169 (0%) | Action a Action-adventure, Adventure, RPG, Puzzle and Mini-game, Educational, Fighting, FPS, Music, Platform, Racing, Simulation, Sports Simulation | N | N | N | Website |
| Mazurek et al. (2015) [97] | 9.79 | USA | 58 (13,8%) | Action-adventure, Action RPG/RPG, Fighting, Music, Platform/Party, Puzzle, Racing, Sandbox, FPS, Simulation, Sports Simulation, Strategy | N | N | Y | Websites |
| McMahon et al. (2013) [98] | NP–CA | Australia | 10 (20%) | Action, Adventure, Puzzle, RPG, Sports, Strategy | N | Y | Y | Combinations–Article reference and authors |
| Mitchell et al. (2015) [99] | 6.135 | USA | 228 (34,5%) | Action, Adventure, Simulation, Educational/traditional | Y | N | Y | Arbitrary: authors |
| Moffat et al. (2017) [100] | 1.206 | USA | 21 (NP) | FPS, Sandbox, Puzzle | N | N | Y | Arbitrary: authors |
| Musetti et al. (2019) [101] | 4.23 | Italy | 366 (15%) | MMORPG, MOBA, Browser, FPS, RTS, Simulation | N | N | N | Arbitrary: authors |
| Ortiz de Gortari and Griffiths (2016) [102] | 4.92 | United Kingdom | 2362 (13,9%) | Action, Adventure, FPS, Racing, Fighting, Puzzle, Music/Dance, Educational, MMORPG, RPG, Simulation, Strategy, Sport | N | N | N | Article reference |

(*Continued*)

**Table 1.** (Continued)

| Authors | Impact factor | Country | Participants (% of women) | Genres | Subgenres | Definition | Examples | Source of taxonomy |
|---|---|---|---|---|---|---|---|---|
| Ortiz de Gortari et al. (2016) [31] | 9.79 | International | 2362 (13,9%) | Action, Adventure, FPS, Racing, Fighting, Puzzle, Music/Dance, Educational, MMORPG, RPG, Simulation, Strategy, Sport | N | N | N | Arbitrary: not specified |
| Palomba (2019) [103] | 2.68 | USA | 30 (40%) | Action, Fighting, Music, Party, Platformer, Racing, Simulation, Sports | N | N | N | Website |
| Payne et al. (2017) [45] | 9.79 | USA | 350 (20,29%) | Action, Adventure, Puzzle, Sports, FPS, MOBA, RTS, MMO, Platformer, Simulator | N | N | Only MOBA | Arbitrary: not specified |
| Peever er al. (2012) [104] | NP–CA | Australia | 466 (18%) | Action Adventure, Action RPG, Board or Card Games, Casual, Education, Fighting, Flight, MMORPG, Music, Party, Platformer, Puzzle, Racing, RTS, RPG, Shooters, Simulation, Sports, Text Adventure and TBS | N | N | N | Arbitrary: authors |
| Potard et al. (2020) [30] | 2.289 | France | 546 (21,2%) | FPS, action-adventure, Sports & racing, RPG, Fighting, Strategy, Puzzle, MMORPG | N | N | N | Article reference |
| Prena and Sherry (2018) [105] | 2.46 | USA | 30 (21%) | Action/Adventure, Arcade, Classic Board Game, Educational, Exergames, Fantasy/Role-Play, Fighter, Puzzle, Racing/Speed, Shooter, Simulation, Sports, Strategy | N | Y | Y | Combination: Article reference and Websites |
| Prevratil et al (2022) [106] | 1.97 | USA | 78 (0%) | Action video games | FPS, TPS, Action RPG, Sport, Driving, Adventure | N | N | Article reference |
| Quiroga et al. (2019) [107] | 2.77 | Spain | 134 (78,4%) | Shooters, Sports, Platform, Strategy, Puzzle | N | N | Y | Combination: Article reference and Websites |
| Ream et al. (2013) [108] | 12.60 | USA | 692 (34%) | MMORPG, FPS, Other RPG, Other Shooter, RTS, Other Strategy, team Sports Simulation, Motion Control Category, Fighting, Driving, Platformer, Rhythm, Other | N | Y | N | Website |
| Rehbein et al. (2016) [109] | 9.79 | Germany | 3073 (52,97%) | Brain & skill, Sports, Racing, Simulation, Actin-adventure, Shooter, Strategy, RPG, No allocation to genre possible | Y | N | Y | Combination: Article reference and German Entertainment Software Self-Regulation Body USK |
| Rodio and Bastien (2013) [110] | NP–CA | France | 70 (1,7%) | RTS, MMORPG, FPS | N | N | Y | Market analysis |

*(Continued)*

**Table 1.** (Continued)

| Authors | Impact factor | Country | Participants (% of women) | Genres | Subgenres | Definition | Examples | Source of taxonomy |
|---|---|---|---|---|---|---|---|---|
| Salmon et al. (2017) [111] | 2.68 | Canada/USA | 416 (74%) | Puzzle and strategy, Educational, Action and adventure, Music, fitness & lifestyle, RPG, Sports and racing, Simulation, FPS, Party, Gambling/casino games | N | N | N | Arbitrary: not specified |
| San Nicolas Romera et al. (2018) [112] | 3.06 | Spain | NA | Arcade, Adventure, Shooters, Educational, RTS, Fighting, Survival horror, Platformer, RPG, Puzzle, Simulation, Nonlinear/Sandbox, Mixed-nature genres | Only for Adventure, Fighting and RPG | Y | N | Arbitrary: authors |
| Scharkow et al. (2015) [32] | 9.79 | Germany | 4500 (43,4%) | Strategy, Puzzle, Sport, Adventure, RPG, Platform, Simulation, Music, action | N | N | N | Article references |
| Seok and DeCosta (2012) [113] | 9.79 | South Korea | 1500 (14%) | RPG, FPS, MUD, RTS | N | N | N | Arbitrary: participants |
| Shliakhovchuk et al (2021) [114] | 3.921 | Spain, Ukraine | 427 (24%) | Other, MOBA, Shooter, Construction, Puzzle, Racing, Sport, Simulation, Strategy, Adventure, Role Playing, Action | N | N | N | Arbitrary: not specified |
| Sjöblom et al. (2017) [115] | 9.79 | Finland | 1091 (7,7%) | Action, CCG (collectible card games), Fighting, FPS, MMO, MOBA, Rhythm, RPG, RTS, Sandbox, Sports | N | N | N | Combination: Article reference and Website |
| Stopf al. (2015) [116] | 3.95 | Germany | 2891 (16,3%) | Action, Strategy, RPG, Simulation, Unclassified | N | N | N | Combination: Article reference and authors |
| Strojny et al. (2023) [117] | 4.7 | Poland | 205 (64.9%) | Action-adventure, MOBA, Shooter, Card games, Strategy, RPG, Sport, Simulation, Puzzle/Logical, MMORPG, Driving, Battle Royale, Survival Horror, Platform, Fighting, Other | N | N | N | Article reference |
| Subramaniam et al. (2016) [118] | 8.713 | Singapore | 1251 (44.5%) | MMORPG, RTS, FPS, Other | N | Y | Y | Article reference |
| Thorne et al. (2014) [119] | 3.84 | Australia | 320 (0%) | Action/adventure, sports/racing, strategy, RPG | N | Y | Y | Website |
| Upton et al. (2022) [120] | 3.36 | USA | 473 (NP) | Action, Action-adventure, Adventure, RPG, Simulation, Strategy, Sports, MMO, Casual/Mobile/Idle, Party, Logic, Casino, Board, Trivia | N | Y | Y | Article reference |
| Vahlo and Karhulahti (2020) [121] | 4.87 | International | 813 (40,6%) | Action, Action-adventure, Adventure, Puzzle, Platformers, Racing, RPG, Simulation, Strategy | N | N | N | Arbitrary: not specified |

(*Continued*)

**Table 1.** (Continued)

| Authors | Impact factor | Country | Participants (% of women) | Genres | Subgenres | Definition | Examples | Source of taxonomy |
|---|---|---|---|---|---|---|---|---|
| Vargas-Iglesias (2020) [122] | 2.18 | Spain | NA | Beat'em Up, Fighting, Flight Simulator, Pinball, Platformer, Shoot'em Up, Shooter, Dating Sim, First Person Puzzle, Graphic Adventure, Interactive Fiction, Interactive Movie, Tile-matching, Visual Novel, Card Battle, Gambling, Grand Strategy, Management, Turn-Based Strategy, Turn-Based Tactics, Life Simulator, Real-Time RPG, Real-Time MMORPG, Sandbox, Turn-Based RPG, Turn-Based MMORPG, Action-Adventure, Action-Puzzle, Maze Game, Real-Time Tactics, Survival Sandbox, Tactical Shooter, 4X/Built and Battle, Real-Time Strategy, Stealth, Tower Defense, ARPG, Racing Simulator, RT-MMORPG, Sports, Free Roaming, Survival Horror, MOBA | N | N | N | Combination: Websites, Game Platforms and Article references |
| Ventura et al. (2012) [123] | 11.18 | USA | 319 (48,6%) | Fighting, RPG, Action-adventure, Puzzle, Social Media, Platformer, Strategy, Simulation, Shooter | N | N | Y | Arbitrary: authors |
| Vermeulen et al. (2017) [124] | 8.957 | Belgium | 464 (100%) | Action-adventure, Strategy e MOBA, MMORPGs, RPGs, Casual e social network games, Music e movement games, Sport games, Fighting games, Platform games, Race games, Shooters, Simulators, Building e resource games | N | N | N | Arbitrary: authors |
| Von der Heiden et al. (2019) [125] | 4.23 | Germany | 2734 (13,1%) | Simulation, Strategy, Action, RPG, Unclassified | Y | Y | Y | Article reference |
| Wohn and Lee (2013) [126] | 2.68 | USA | 164 ("about 60%") | Social network games (sNGs) | Farm Sim, Arcade, Pet Sim, RPG, Restaurant Sim, Word Games, Brain Games, Card Games, Town Sim | Y | N | Arbitrary: not specified |
| Yang et al. (2022) [127] | 4.23 | Canada | 351 (59%) | Puzzle, action, Platformer, RPG, Strategy, Sports, Construction and management, Social Simulation, Idle, Other | N | N | Y | Arbitrary: authors |
| Yang et al. (2023) [128] | 9.14 | USA | 367 (28.1%) | FPS (First-Person Shooter) Games, MMORPG (Massively Multiplayer Online Role Play Video Games), MOBA (Multiplayer Online Battle Arena) Games, Fighting Games And Sports Games | N | N | N | Arbitrary: authors |

(*Continued*)

**Table 1.** (Continued)

| Authors | Impact factor | Country | Participants (% of women) | Genres | Subgenres | Definition | Examples | Source of taxonomy |
|---|---|---|---|---|---|---|---|---|
| Yilmaz et al. (2022a) [129] | 11.18 | International | 523 (52%) | Genres by scenarios: Action, Action-Adventure, RPG, Simulation, Strategy, Racing/driving, Sport, Puzzle Genres by primary production purpose: Entertainment, Educational, Serious Traditional game genres: RPG, Sport, Race, Rope, Action, Fight, Puzzle, Strategy-Intelligent, Board | N | N | N | Article reference |
| Yilmaz et al. (2022b) [130] | ? | International | 510 (63%) | Genres by scenarios: Action, Action-Adventure, RPG, Simulation, Strategy, Race, Sport, Puzzle Genres by primary production purpose: Entertainment, Educational, Serious, Casual, Electronic sport Traditional game genres: Role-playing Animation, Sport, Race, Rope, Action, Fighting, Lego—Puzzle, Strategy, Stone, Board | N | N | N | Article reference |

Note. NP–CA–Not provided–article from conference, NA—not applicable, the subject of the analysis was data collected without direct human participation, Y–yes, N–no.

### 3.1 Quality of creating game taxonomies

Gaming taxonomies distinguished by the authors of articles included in this systematic review have multiple different sources. During our review, we identified 46 unique taxonomies in 96 papers. It means that approximately 48% of taxonomies were used only once. Moreover, only in the case of 50 papers (52%) authors utilized some kind of external or combined source of taxonomy. The specific numbers are shown in Fig 2 below.

Unfortunately, the way of constructing game genre classification was not specified in eighteen (19%) articles [31,45,46,48,58,60–63,67–69,74,76,111,114,121,126]. It can be assumed that the construction process was arbitrary. On the other hand, in twenty-three articles, the classification was based on the assessment of the authors and/or was study-specific [19,20,25,27,51,56,64,70,72,75,78,82,87,93,99–101,104,112,123,124,127,128]. Furthermore, the classification that was based solely on the judgments of the participants appeared in three articles [80,83,113]. Mathews et al. [95] created a game genre taxonomy by using expert judges. Moreover, some researchers used the following websites to gain information about game genre: Gamefaqs.com, Pegi database, Naver database, Gamespot.com IGN.com, Amazon.de, Amazon.jp [33,44,53,59,77,84,96,97,103,108,119]. Except for websites, gaming platforms such as Steam were also used to create game classifications [50,54,73,89]. Gilbert et al. [71] and Rodio & Bastien [110] built their classification by analyzing the market sales in 2010. Some taxonomies (n = 19) were distinguished based on previous literature [29,32,49,52,54,57,64,90–92,102,105,106,117,118,120,125,129,130]. Articles used as references for other taxonomies were as follows: Lucas and Sherry [131], Apperley [43], Zammito et al. [132], Adams [133], Lee

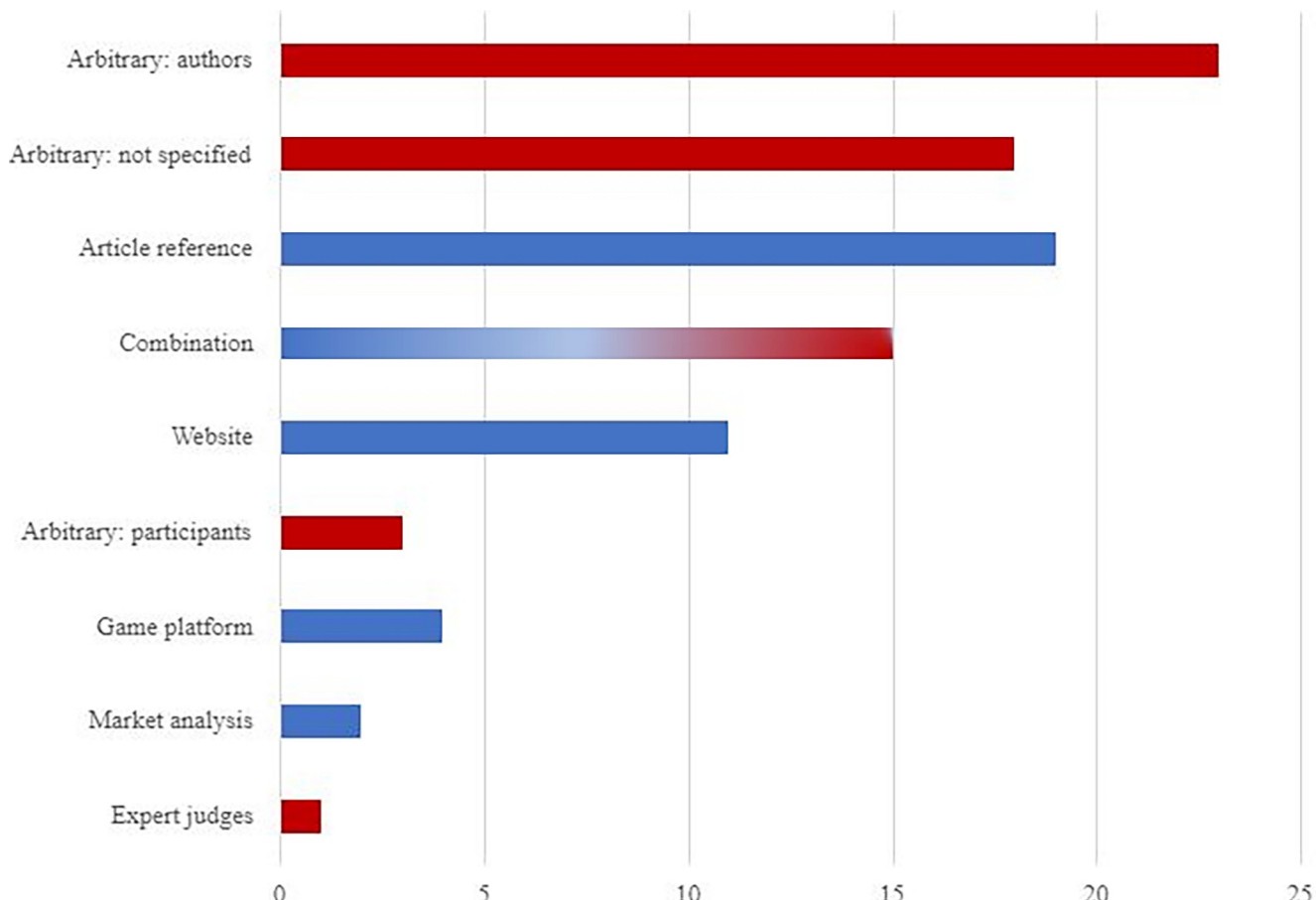

**Fig 2. The number of empirical articles depending on the source of the game taxonomy used in them.** *Notes*. Red bars represent study-specific taxonomies (including arbitrary); blue bars represent taxonomies based on external, publicly available, sources.

et al. [85], Lemmens and Henriks [86], Rehbein et al. [109], Dale and Green [134], Green et al. [72]. Other researchers (n = 12) used various combinations of the abovementioned sources. [30,55,65,79,81,85,86,88,94,98,107,109,115,116,122]; out of these 15 cases, in 12 authors combined previous articles with other sources. The way in which game taxonomies are constructed seems not to be dependent on the journal's impact factor. However, it is worth mentioning that in journals which impact factor is above 11, authors sourced their game classification on a website, Gamefaqs.com, and also provided definitions for each single genre [66,108]. It is important to mention, thatYilmaz, Yel and Griffiths (129,130) presented three video game taxonomies. The first was focused on game's scenarios, second primary production purposes and last one presented traditional game genres.

It is also important to mention that only twenty-six articles not only listed several game genres included in their research but also defined them [20,21,44,46,51,53,57,66,71,74,76,78,81,85,89,94,95,98,105,108,112,118–120,125,126]. However, some provided definitions are not separable; e.g., Thorne [119] used "puzzle-solving" phrase to describe both Action/adventure and strategy genres. Examples of games that were categorized into specific genres were provided in almost half of the articles included in the systematic review (n = 37). Usually, the article which contains the definition also presents the examples of games characteristic for the genre, however, the authors of twenty-two papers did

not mention neither definition nor examples. The lack or presence of definition and examples of games for various genres seems not to be related to the impact factor of the scientific journal.

## 3.2 General characteristics of taxonomies

The list of genres included in all of the studies is presented above in Table 1 and in S1 Table in Supporting information. It follows from them that the most mentioned genres in all the articles were Sport (n = 75), RPG (n = 75), Strategy (n = 57), Simulation (n = 56) and Puzzle (n = 55), Driving/Racing (n = 50) MMORPG (n = 44) and FPS (n = 43). As was already mentioned, Sport was the most mentioned genre in the studies included in this systematic review. However, it is important to emphasize that multiple studies not only include broad Sport category but also present multiple sport disciplines not as subcategories but as different game genres such as Driving/Racing (n = 50) or Fighting (n = 43). The cause of these differentiations may be connected to the mechanical and storyline aspects of these genres which include not only simple simulation of sport disciplines but also elements of fiction such as racing in fantasy lands (e.g., *Mario Kart*) or using inhuman powers in fighting games (e.g., *Inhumans*, *Tekken*). Furthermore, it is important to emphasize that some researchers debate whether Sport games, which are simulation of real-life sport should be differentiated from Simulation genre [57,94]. On the other hand, Simulation as a genre is not only associated with traditional sports; it can also be used to describe video games that focus on management or construction (such as Sim City, Planet Zoo, Railway Empire, and Train Simulation), which simulate real-life events or activities. Rehbein et al. [109] even distinguished five different Simulation's subgenres such as Simulation and construction, Life simulation games, Business simulation games, Sandbox physics games and Flight simulation games.

Aspects of management and construction in Simulation games are strongly related to Strategy games. This genre was mentioned 57 times in the articles included in this study. However, it is important to emphasize that researchers usually diversify its subgenres–RTS (n = 30), TBS (n = 13) and Management (n = 8). Some studies included genres that combine Strategy and RPG elements such as Turn based RPG [52,88,121]. MOBA, one of the most popular game genres among players, was mentioned only 27 times, mostly in articles published after 2015. Similarly to Strategy genre, Action (n = 39) also has been usually differentiated into Action-adventure (n = 39). Moreover, this genre is also represented in such multigenres as Action RPG (n = 8) [52,61,72,79,82,88,104,121] or Action-Puzzle [74]. López-Fernández et al. [92] also included Action-shooter as an individual genre without including Shooter or FPS category in their study.

Action games in the past were usually defined as shooters [4], so creating new genre may create unnecessary difficulties in defining genres. Shooter as a genre was included in 33 articles. However, it is important to mention that in most studies where researchers included Shooters in their genre classification, they did not add FPS to it [25,29,55,80,88,89,104,107,108,112]. Furthermore, FPS was included as a subgenre of Shooters in three studies [19,79,109]. Only three studies mentioned TPS (third person shooter) as a subgenre of shooters [19,74,125]. As presented above, FPS has been much more present in the studies than the broader Shooter genre. It implies that first-person shooter started to be used as a replaceable name for shooters even though it is a much narrower category as a subgenre that cannot include TPS in it. Other genres related to Shooters included in the studies were Arcade/Shooting [81], Hero-shooter which is defined as "MOBA inspired first person shooter" [76], Battle Royale [19,76,87,117], Beat 'em' up [61,63,84,93,122] and Top-down shooters [88] which are also known as Shoot 'em' up [109].

Recently, new mechanics and multigenres such as Adventure RPG [120] and Text Adventure [74,104] were diversified from the broad category of Adventure games (n = 30). Furthermore, some researchers mention Survival and Horror games as an entirely excluded category where player need to survive in the potentially scary and extreme environment [61,74,88,95,112,117,122]. However, some researchers included such games in the Adventure and Thrillers category [94].

As was mentioned before, Puzzle games were presented as an individual genre in 55 articles. Puzzle games are defined as "games involving matching, logic, deductive reasoning and other" [44]. Hazel et al. [74] differentiated three Puzzle genres: Puzzle Action, which combines elements of Puzzle and Action game in which player solves puzzles by using a first- or third-person avatar; Puzzle Other–where solving logical mysteries happens without an avatar and individual category such as Point and Click which is defined as "The player uses a cursor to interact with a flat environment, usually to solve puzzles and experience a story". Such definition of Point and Click suggests occurrence of puzzle and adventure aspects in game. Furthermore, Hazel et al. [74] distinguished Hidden Objects which definition implies inclusion in the Point and Click category: "A form of Point and Click game where players locate items in a busy environment". This category also appears in an article by Rehbein et al. [109]. However, it is presented as a subgenre of a broader category called Brain and Skills, which also included Card Games, Puzzle Games, Board Games, Quiz Games, Skill Games, Fitness Games, Music Games, Party Games and Game Collections. As was said before, definition of Puzzle games usually includes aspect of logic [44]. Despite that, Logic genre was differentiated in two articles as an individual genre [69,120].

On the other hand, Platform Games were mentioned 33 times as a standalone genre. However, Platform games have recently been categorised as a subgenre of Action or Puzzle games, even though they are often understood as one of the first genre of games [135]. Elliot et al. [44] defined platform games as games that involve action of jumping and running (e.g., *Mario Bros*). San Nicolas Romera et al. [112] added avoiding opponents attacks by using some objects or artifacts. It seems important to mention that Jump'n'Run which definition seems to resemble Platform games was mentioned as its subgenre [135]. It was also listed as an Action-adventure subgenre by Rehbein et al. [109] and an unclassified subgenre in an article by von der Heiden et al. [125]. Dieris-Hirche et al. [63] mentioned it as a standalone category. Platform games were also listed as a merged category with Party games by Mazurek et al. [97]. In relation to that, Party as an individual genre was mentioned in seven articles [20,55,56,60,103,104,106,111,120,122]. Furthermore, another gaming genre that can be played individually or with a group of friends is Rhythm/Music games (n = 27). Similarly, recently, Board or Card games have been played with use of specific applications giving the opportunity to play different games (virtual tables) such as the *Tabletop Simulator* or individual games such as *Magic*: *The Gathering Arena*, *Heartstone* or *Legend of Runaterra* (n = 23). Some researchers distinguish Gambling and Casino games from Board/Card games category [44,53,66,95,111].

Another category of games are browser games (n = 9) or games on social media apps (n = 19), which are defined as simple games that could run on a simple platform or social media network [20,126]. It is interesting that Wohn et al. [126] listed some subgenres of social network games (SNG) that usually focus on simulation–farm, pet, town or restaurant simulation. They also differentiated such SNG's subgenres as Card, Brain, Word and Arcade games. Some researchers also include SNG in the casual game genre, alternatively called mobile or idle, in which games are defined as being designed for short periods of play and having a portable nature [27,65,93,104,120]. It is important to mention that category Other for unclassified genre of the games or mixed nature of the game's genre was included in sixteen articles [19,44,58,63,66,78,91,108,109,112,114,116–118,125,127].

### 3.3 One genre, multiple definitions

The present review highlighted the definitional inconsistency of the game genres under study. Due to the lack of convergence in the genre and sub-genre classification of games, it is difficult to clearly state for which of the genres the greatest discrepancy in definitions occurred since the definitions particularly depend on the methods of classification. It is noteworthy that, excluding studies that referred to previous studies in the context of classification, total convergence among the publications subject to this review did not appear for any game genre. There were also numerous instances where no definitions were provided and the genres' classifications were arbitrary [19,25,27,31,32,45,48,56,58,60–64,67–70,72,75,79,80,82,83,87,93,99–101,104,111,113,114,121–124,127,128], thus leaving many unknowns.

One of game-play virtuosity as the gamer controls every move of the counter that usually can kill and can be killed." [57], "Include physical challenges and require the greatest classification discrepancies concerned Action games due to the variety of definitions, classification as falling under/combining with other genres, or the lack of their inclusion. In the studies subject to this review, the Action genre has been defined disparately, for instance: "Games that fell into the action category shared a focus on physical enactment to advance in the game, often using violent or dangerous behaviors." [71], "Action games need skills in hand-eye coordination to complete objectives, which include defeating opponents." [120]. Moreover, some studies have classified the Action-adventure genre as one [19,30,44,48,49,51,54,56,61,64,66,72,79,80,82,83,86,91,95,97,104,105,111,117,119,120,122–124,129,130],some have found them separate [31,32,45,53,55,60,73,93,98,99,102,106,114,115], and some have classified them as both one genre and two separate genres [29,33,52,84,85,96,109,121] drawing further inaccuracies in the context of classifying this genre as deserving a separate distinction or not.

The greatest definitional uniformity was visible in the case of Sport games, which were most often described as simulations of real-life sports (e.g., football, basketball, etc.) [e.g., 20,44,71,76]. However, there were cases where Sport games were treated as part of the Simulation genre [e.g., 57,112,125] or where Sports and Racing games were classified as one genre [e.g., 30,111,119]. There were also two instances where Sport games were classified as a part of the action genre [e.g., 52,99]. The aforementioned examples serve as evidence that even within the genre characterized by the highest degree of classification homogeneity, inaccuracies still persist.

### 3.4 The multi-genre misunderstanding

The general range of game genres proposed by the authors also differed noticeably, regardless of the year of publication, which also contributed to the diverse understanding of the genres. A study published in 2012 [44] lists the following 16 genres: MMORPGs, Other RPGs, Action-adventure, First-person shooter (FPS), Other shooter, Sports general, Sports other, Rhythm, Driving, Platformer, Real-time strategy, Other strategy, Puzzle, Board and card games, Gambling, Other; whereas a study published in later years [32] mentions the following 9 genres: Strategy, Puzzle, Sport, Adventure, Role-playing (RPG), Platform, Simulation, Music, Action. Surprisingly, a study published a year later [57] proposes only 4 genres: Action, Role-playing, Simulation, Strategy; and another study [103] proposes 8 genres: Action, Fighting, Music, Party, Platformer, Racing, Simulation, Sports.

The presented classifications of game genres are surprising, considering the fact that the gaming market is constantly expanding, offering new genres that are fusions of other previously existing genres (multi-genres), for instance, MMORPGs. Multi-genre game developers extract the immersive mechanics of two or more game genres and combine them into one product to achieve higher player engagement. This phenomenon was highlighted in the case of

MMORPGs, which, as demonstrated in the introductory part of the present review, turn out to be one of the most associated with Gaming Disorder. Therefore, in order to get to the essence of this phenomenon by comparing research results and building a coherent set of information on this subject, the uniformity of game genres in research is substantial. Yet, according to the classifications proposed in the mentioned studies, MMORPGs could be classified as MMORPGs themselves but also as Role-playing and Action simultaneously. This way of classifying becomes problematic given the fact that the Action game genre appears in each of the classifications and only one of them recognizes MMORPGs as such, leading to confusion not only in the classification of MMORPGs but also in the understanding of other genres.

### 3.5 Steam tags–different approach to game classification

Li and Zhang [89] proposed that game genres, instead of strict taxonomies, should be based on the more flexible approach offered by Wittgenstein's "family resemblance" theory. Researchers tried to construct their gaming classification by collecting and analyzing user-generated tags from the Steam gaming platform. Results of calculated weighted degree, closeness centrality, betweenness centrality, and PageRank show that five game tags obtained the highest scores in these metrics. They were as follows: Indie, Action, Adventure, Singleplayer, and Casual. Furthermore, four game genres were distinguished on the basis of "high-centrality gameplay-designing tags within the model Video Game Metadata Schema category and Steam categorization" [89]. These major game genres were Strategy and Simulation Games (Game Tags' examples: "Strategy", "Simulation", "Multiplayer", "Survival", "Sport", "Medieval", "Turn-based") Puzzle and Arcade Games (Game Tags' examples: "2D" "Great Soundtrack", "Puzzle", "Comedy". "Arcade", "Platformer"), RPG Games (Game Tags' examples: "RPG". "Story Rich". "Fantasy", "Choices matter", "Hack and Slash", "Visual Novel", "Point and Click", "Text-based") and Shooter Games (Game Tags' examples: "Atmospheric", "Sci-fi", "Shooter", "Exploration", "FPS", "Horror", "TPS", "Violent"). Such a way of presenting genres includes major story themes, mechanics or subgenres, ratings, and atmosphere/mood.

Furthermore, Li [88] in his next study conducted Exploratory Factor Analysis (EFA) on game tags from 22 749 video games on the Steam platform. Results of EFA implied the presence of 29 game genre factors: Arcade Shooter, Fighting, Soccer, Sandbox, TBT, Resources Management, Music, Shooter, Rogue, Character Action, Strategy, Classic, Board/Card, Gal Game, Platformer, Survival, MMO, Racing, RPG, Strategy RPG, ARPG (Action RPG), Top-down Shooter, Interactive Fiction, Tower Defence, RTS, Simulation, Exploration, Parkour and Education. Full table with correlated Gameplay Tags can be found in Table 1, page 212 [88]. Even though twenty-nine genres were distinguished by EFA it still did not create totally separable game genres. Furthermore, Li [88] also noticed that the new generation of video games is multigenre by nature. In turn, analyzed games belonged to one single genre or a combination of several genres. Li suggested that each game can be characterized by a certain list of game genre factors; for example, Divinity Original Sin 2 can be described as "Turn Based Tactics" and "RPG". Using game factors may not only facilitate customer needs but also be important for scientific reasons.

## 4. Discussion

The aim of this study was to explore the ways of defining video game genres in psychological and computer science research. The review was conducted without a prepared protocol. The results of this systematic review expose difficulties in creating classifications of video games. First of all, scientific research has a hard time keeping up with the fast-paced evolution of gaming market. New genres, mechanics, and themes have been constantly emerging, which makes

it difficult to preserve distinct genres [5]. Consequently, it is far more challenging to create a constant definition of video game genres and conduct long-term comparative research about their effects. Another problem that is highly related to the last one is the tendency of developing games that could be categorized into multiple genres [11]. A perfect example of this phenomenon is the creation of such genres as Action RPG or MOBA–which are hybrids of Real time strategy and Action genre. Overlapping genres may create difficulties in analyzing effects of genre on, for e.g., Gaming Disorder or cognitive functions. Therefore, it can blur the image of how a specific genre may be characterized as more "addictive" or how it can improve an individual's executive functions. Another important factor is the complexity of mechanics, themes, and aesthetics of new games [9,85]. Focusing solely on the game genres may oversimplify the effects of their several components on psychological functions, which can potentially lead to biased or incomplete findings. Furthermore, this systematic research shows the lack of consistent standardization of game genres [9]. Many researchers create their classifications on the basis of subjective criteria that may be different for each individual. Consequently, researchers categorize the same game into totally different genres. Some define one genre as a distinct genre, and others present it as a subgenre. Such a situation creates confusion and makes comparative research virtually impossible. Furthermore, focusing only on the impact of a genre in psychological research may lead to its over-pathologizing, as it is with the relationship between MMORPG and Gaming Disorder [136]. However, as it is known, it is not the overall genre that creates risk of addiction but its structural elements such as mechanics, social aspects, no define ending, identification with the gamer's character, rivalry, daily missions or time-limited events, etc. [137]. The abovementioned reasons lead to the question: are game genres really sufficient tool to be used in scientific research? Researchers seem to have two possible directions–we can create another as complete and up-to-date as possible video game genre taxonomy, which will probably not stand the test of time in the rapid evolution of the game industry, or we can try to focus more on specific gaming mechanics, atmosphere, themes, and aesthetics, which can be found by analyzing universal game tags on gaming platforms such as Steam, GOG, Epic Games, Ubisoft Connect, or EA [89]. We have decided to pursue both of these directions in order to provide future researchers with a comparison of both options and, consequently, an opportunity to make a decision that best suits their research goals.

## 4.1 Yet another old chestnut

This systematic review summarizes video game classifications used in psychological research between 2012 and 2023. We propose to create a game genre classification that consists of all the video game genres mentioned in the coded articles. As presented in S1 Table in Supporting information, the video game genres listed from the most mentioned are as follows: Sport, RPG, Strategy, Simulation, Puzzle, Driving/Racing, MMORPG, FPS, Fighting, Action-adventure, Action, Platform, Shooter, Rhythm/Music, MOBA, Arcade, TBS, Educational, Browser, Party, Management, Action RPG, Survival/Horror, Sandbox, Beat 'em' up.p. Genres mentioned less than five times are recorded in the section Other, e.g., Gambling, TPS, Retro, Logic, Battle Royale, Hero-Shooting, Quiz/Trivia, Roguelike etc. This categorization presents over a decade of research regarding using video games genres and it can be useful material for further studies if we decide to continue using video game genres. However, it must be remembered that due to the fast-paced development of the gaming industry and the creation of new game genres that combine the mechanics and characteristics of two or more genres, it may not stand the test of time in the long run.

## 4.2 New approach

The essence of psychological research is to understand and explain human behavior, emotions, and cognitive processes. It is this focus on understanding their mechanisms at the individual, interpersonal, and intergroup levels that distinguishes psychology from other disciplines that may study the same phenomena but take different perspectives. With this in mind, the question arises: "What is the purpose of investigating games' characteristics in the context of psychology?". There is no doubt that the first researchers dealing with this topic set themselves the goal of understanding the relationship between psychological phenomena and the characteristics of games. For instance, McClure and Mears [138] attempted to answer the question of the relationship between 'intellectual efficiency' and video game genre preferences. They report the following discovery: "Brighter people (. . .), liked simulation and adventure games more than arcade games" (p. 274). The authors do not explore this discovery in the discussion of the results, but a certain distinction is striking—simulation and adventure games seem to be more complex than arcade games. It is worth noting that this distinction may have been useful in 1984, a year when around 400 video games were released worldwide–Wikipedia lists 393 titles to be exact [139]. In those days, distinguishing between three genres and leaving the result uncommented could be justified. Since then, the gaming market has grown incredibly. There have been more games and, consequently, labels that can be used to categorize them. Suffice it to say that Wolf [7] identified as many as 42 game categories. In our list of genres presented above, we limited ourselves to the 25 most popular genres, which is still an impressive number. In 2023, 13 346games were released on one platform (Steam) alone [140]. Therefore, even more questions arise: "Can the numerous game genres used by game producers and consumers be informative for a psychologist? What is behind these distinctions? Can we, as in 1984, reliably map the genres to more general characteristics of the games?". Numerous studies have shown that MMORPGs appear to be associated with a greater risk of problematic gaming [136]. Is there a practical possibility that, based on these observations, actions that will protect users from gaming disorder will be taken to target MMORPGs as a genre? This seems unlikely for various reasons. One of the main arguments against such an action is that it is not the genre that makes the game 'dangerous', but its features, mechanics, setting, promotion and finally the monetization model. It is worth noting that in some areas, from the very beginning, the approach from the perspective of game features rather than their genres dominates. For example, Silvern and Williamson [141], in one of the first papers on the relationship between games and aggression, write about 'violent games' instead of writing categories based on genres. So why not focus the search for game-located determinants of positive (e.g., cognitive development) and negative (e.g., gaming disorder) consequences of gaming on factors of psychological meaning rather than labels designed for game development and cultural studies? We postulate that the tags created by the gaming community can be an interesting alternative for genres.

## 4.3 A glimpse of hope

In this section, we will present the mechanism of operation of community-based tags and the possible benefits and limitations of their use as an alternative to genres. We'll use Steam labels as an example (so-called 'Steam tags'), but many other game-selling platforms offer similar solutions that could potentially be used (e.g., GOG, Epic Games, Ubisoft Connect, or EA) [89]. Steam tags are descriptive labels assigned to games on the Steam platform. They are utilized as a way to classify games based on different aspects such as genre, gameplay features, themes, and visual properties. Steam tags are intended to improve the sale of games, with the assumption that by processing them, it becomes possible to better match the individually offered

games to the needs of the recipient, which positively affects the willingness to buy [142]. Tags can be assigned by both developers and users; hence, every individual interacting with a game is able to shape its overall image through tags. The tags cover a wide range of characteristics, including genres. Some common examples include "action", "strategy" and "RPG". However, tags can also be more specific, capturing unique aspects of a game, such as "idle", "trading card game", "co-op", "loot", " replay value", or even "addictive". There are over 420 default Steam tags (including 79 from "genre " and "61 " from "sub-genre " category), yet the list of tags is not closed, and each user can propose any tag that will be assigned to the game after approval by the moderators [142]. The number of tags assigned to a given title is unlimited, but as a result of the activity of users and publishers, the most important ones are placed first. Tags proved to be an effective way to describe games, they were introduced to Steam in 2014 and have been around ever since. It is worth noting that on this platform there is no longer a division of games by genre.

Obviously, tags (like genres) were not designed as a tool for psychological researchers; therefore, the mere fact of their existence and popularity is not a sufficient reason to depart from the traditional classification of games. However, they have several advantages over genres that may lead to interesting research results. First of all, the tags assigned to games are the result of a dynamic consensus among their developers and the gaming community. Furthermore, they are created by the community, so they reflect how players perceive and categorize games, which can give an important insight into players experiences. The tags are ordered on the game page on the basis of their popularity, which allows for easy estimation of how crucial a given characteristic is in the overall picture of a given game. At this point, however, it should be noted that the publisher can remove tags, which may make interpretation uncertain to some extent. Using steam tags may offer access to the large sample dataset, which not only covers genre but also may emphasize so-called facet characteristics of games such as gameplay, style, purpose, theme, target audience, temporal aspect, setting, and mood/affect which can be used in psychological research to distinguish individual structure characteristics that may elevate risk of gaming disorder [85,89]. Consequently, we can avoid the overpathologization of some game genres and focus on individual mechanics that may increase or decrease the risk of the potential development of behavioral addiction. Moreover, data from gaming platforms can be obtained much faster than individual and subjective categorization of games into the respective game genre. It can even be automatized. One of the main findings of this systematic review was that genre classifications are very subjective. We can also say that about game tags. However, in the first situation, a few researchers usually decide that this game will be classified as, for example, RPG. In the second situation, game developers and a large sample of the gaming community decide on tags for each game. External databases such as SteamSpy (https://steamspy.com/) provide data about how many users assigned each individual tag to an individual game. On the basis of this data, we can obtain quantitative indicators that can be further analyzed. Furthermore, using tags from worldwide accessible gaming platforms could create a way to conduct comparative research.

Even though game tags have many advantages, we still need to remember about some of the risks of using them in scientific research. First of all, game tags on each gaming platform can be diverse from each other, so relying on game tags from one specific platform may be limited. An aggregation of available data may be helpful in this case. Secondly, game tags may not present all the valuable characteristics of video games, which may have an effect on the individual's psychological functioning. Perhaps, using more structured metadata repository like, for example, Gameopedia could be a way to cover all the important mechanics from a psychological point of view.

## 4.4 Gender discrepancy

This systematic review presents research over one decade and it is surprising to see that only 26% of all the participants in the studies were women. According to recent data from the Entertainment Software Association [143], 48% of players identify as women. It is a huge discrepancy between the reality presented in scientific research and the real gaming community. Such a situation may be a result of stereotypes that playing video games is primarily a male activity or the untrue belief that women play only casual games [144]. This narrow focus excludes and overlooks the experiences of women gamers, leading to a lack of representation in the research literature. On the other hand, researchers might face challenges with recruiting women participants because they may be situated outside of video gaming culture; e.g., women are engaging less with the gaming community, which has a history of being unwelcoming and discriminating towards them [144,145]. However, conducting studies on female gamers regarding their gaming preferences, risks of gaming disorder, and gaming motives is very important. Especially when recent findings show that women have very similar risk of developing gaming disorder to male gamers and that they tend to invest more time in playing video games than men [18,146].

## 4.5. Geographical coverage

The geographical distribution on gaming genre research seems to be heterogeneous. The most studies have been conducted in Europe (41%), which compares with different continents (Asia– 12%; Australia and Oceania– 8%; North America- 28%) generates some imbalance. Interestingly, in Asia identified only 12 studies, since this region has the largest population of players [147], so it would be expected that it will be dominant in terms of scientific research. Then, no studies have been identified in South America and Africa. Interest in game genres is not universal on all continents, therefore it can be assumed that it depends on culture variation and technological development.

## 5. Limitations

The review contains several limitations that need to be addressed. Firstly, the paper was not intended to include theoretical and debate articles on the topic of game genres. In view of the fact that the main focus was on providing the most prominent methods of classifying video game genres, the weight of evidence (for example, effect sizes) and methodological quality were not evaluated. Additionally, it was decided to limit the time period of published articles to 2012–2023. The choice was made due to the purpose of presenting an updated view of over a decade of gaming classification in research practice. There is a risk that some valuable publications were overlooked because of their previous date of publication. Perhaps it explains the fact that as many as twenty-two authors did not provide the genres' definitions. Many game genres, like "Strategy", "Action" or "Educational" games have been established since 1980 [6], and defining them may be considered unnecessary. Finally, it should be noted that the selected languages of articles may increase the risk that relevant, non-English, and non-Polish papers will be missed. Moreover, most of the included studies were conducted in Europe ($n = 40$), which indicates an imbalance compared to different parts of the world and probably an imbalance of used languages. Then, the large gamer population is based in Asia; estimations for 2023 provide a number of 1.48 billion [147], so it could be expected that Asian researchers may publish their articles in other languages than English. However, the review includes also studies carried out in Asia ($n = 12$), North America ($n = 27$), Australia, and Oceania ($n = 8$), so we can assume that the potential limitation of strong geographical bias was missed.

## 6. Conclusions

The original aim of this study was to create an up-to-date picture of the taxonomy of game genres used by researchers in recent years. This led to the proposal of a comprehensive taxonomy. Before starting the work, we assumed that we would encourage authors of future studies to use our taxonomy. The main motivation in this case was the possibility of making comparisons between studies, cultures, and years thanks to the use of a unified taxonomy and the facilitation of the research process thanks to the possibility of using a ready-made taxonomy. However, in the course of the review, we realized that, at least for some research applications, attempts to use game genres as proxies of psychologically relevant characteristics may be obsolete. The state of psychological research with the use of game genres that we have found is worrying. We identified problems related to contradictions in genre definitions, the risk of classifying the same games into different genres, and the emergence of new genres, in particular hybrids of earlier ones. Also, the very process of formulating the taxonomy left room for discussion in most cases. In a word, it seems that researchers, creating new taxonomies, are trying to keep up with the development and changes of the market, but they still have not freed themselves from the habits straight from the nineties. As a consequence, we decided to limit ourselves to presenting a list of the 25 most popular genres that have appeared in research over the span of the last eleven years. We supplemented it with a complete list of all genres that occurred in one to four studies (see S1 Table). We believe that, in some cases, the use of this taxonomy will prove useful to researchers.

However, in psychological research, we are obviously interested in psychological phenomena, so why focus on genres when certain mechanics, setting, or other characteristics may be responsible for the occurrence of certain psychological effects? Genres can be treated as constellations of lower-order characteristics; for example, it's hard to imagine an MMORPG without multiplayer, but do only MMORPGs offer multiplayer? If we ask ourselves a psychological question, for example, "Can the pressure resulting from the self-presentation motivation in front of fellow players act as a risk factor for gaming disorder?", we should not divide games into MMORPGs and non-MMORPGs but ask what mode the participants of the study play in, or at least what modes the game offers. There are many indications that this obvious statement may finally be put into practice. As demonstrated by Li and Zhang [89], it is possible to differentiate the characteristics of games without the genres that have so far acted as proxies. Moreover, as Li [88] showed, the use of statistical methods leads to the separation of new categories that may be more useful. Therefore, we wholeheartedly recommend that authors of future research rely on game characterization tags rather than rough classifications.

## Supporting information

**S1 Table. A table presenting in detail the genres used in individual papers.**
(DOCX)

## Author Contributions

**Conceptualization:** Jolanta Starosta, Paweł Strojny.

**Data curation:** Jolanta Starosta, Patrycja Kiszka.

**Funding acquisition:** Paweł Strojny.

**Investigation:** Jolanta Starosta, Patrycja Kiszka.

**Methodology:** Jolanta Starosta.

**Project administration:** Jolanta Starosta, Paweł Strojny.

**Visualization:** Jolanta Starosta, Patrycja Kiszka, Paweł Strojny.

**Writing – original draft:** Jolanta Starosta, Patrycja Kiszka, Paweł Strojny.

**Writing – review & editing:** Jolanta Starosta, Patrycja Kiszka, Paulina Daria Szyszka, Sylwia Starzec, Paweł Strojny.

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
