## [Decision Letter · Decision Letter 0]

2 Nov 2023

PONE-D-23-27687The tangled ways to classify games: A systematic review of how games are classified in psychological researchPLOS ONE

Dear Dr. Kiszka,

Thank you for submitting your manuscript to PLOS ONE. After careful consideration, we feel that it has merit but does not fully meet PLOS ONE’s publication criteria as it currently stands. Therefore, we invite you to submit a revised version of the manuscript that addresses the points raised during the review process.

We look forward to receiving your revised manuscript.

Kind regards,

Patrick Charland

Academic Editor

PLOS ONE

Journal Requirements:

**Additional Editor Comments:**

Dear Dr Kiszka,

the two reviewers have made their decision and changes are requested in your paper (see below).

In addition, I invite you to review the submission guidelines and particularly the reference style section: PlosOne requires references in Vancouver format.

https://journals.plos.org/plosone/s/submission-guidelines

Reviewers' comments:

Reviewer's Responses to Questions

**Comments to the Author**

1. Is the manuscript technically sound, and do the data support the conclusions?

Reviewer #1: Yes

Reviewer #2: Yes

2. Has the statistical analysis been performed appropriately and rigorously? 

Reviewer #1: Yes

Reviewer #2: N/A

3. Have the authors made all data underlying the findings in their manuscript fully available?

Reviewer #1: Yes

Reviewer #2: Yes

4. Is the manuscript presented in an intelligible fashion and written in standard English?

Reviewer #1: Yes

Reviewer #2: Yes

5. Review Comments to the Author

Reviewer #1: This nicely written manuscript aims to present a systematic review of the available English or Polish literature on the classification of video game genres in the context of psychological research with the rapid evolution of the gaming market, based on a systematic search of 4 databases(ScienceDirect, Taylor and Francis, Scopus and PubMed ) Authors propose a new classification of video game genres based on the review of 62 relevant papers and advocate a more detailed understanding by focusing on specific gaming mechanics, and thus introduce the innovative concept of utilizing community-based tags, such as Steam tags, as an alternative to genres in psychological research. Authors should be commended for their efforts. Please consider the following:

1. Was the protocol for this systematic review preregistered( on any site like PROSPERO) or pre-published? Please provide details.

2. Detailed search strategy for at lease one database may be provided.

3. The methods section says "The first reviewer screened ScienceDirect and Taylor and Francis records, whereas the second reviewer focused on Scopus and PubMed records." Usually all databases are searched by two reviewers and articles are also selected by both of them, and any discrepancy is resolved by third reviewer.

The manuscript is actually silent on whether screening and selection was done by two reviewers.

3.

Reviewer #2: It is seen that the article titled "The Tangled Ways to Classify Games: A Systematic Review of How Games are Classified in Psychological Research" is an original study and touches on an important problem that exists in the literature. However, The following major revisions need to be fulfilled to publish the study.

1. I wonder why the researchers did not include the terms "digital game" and the adjacent "videogame" in their search strategies. Because although the term "video game" is predominantly preferred, we also encounter the use of "digital game" and "videogame" in the literature. This not only leads to the exclusion of studies that meet the criteria but also reduces the content validity of the research. For example, Is there any possibility that the study below was not encountered for this reason and could not be included in the manuscript? Because it seems, the videogames were categorized by their genres as well as traditional games in the study.

"Yılmaz, E., Yel, S., & Griffiths, M. D. (2022). Comparison of children's social problem-solving skills who play videogames and traditional games: A cross-cultural study. Computers & Education, 187, 104548.

2. It appears that the databases were searched separately by two different researchers and that a single researcher decided on the studies to be included. There is a problem here in terms of coder reliability. Each database should have been screened by at least two researchers in case of an oversight or indecision, and the procedure followed in case of disagreements should have been explained.

3. The researchers need to add not only years but also months to the period (i.e. January 2012 to December 2022 (inclusive)).

4. The results are given within a certain logical framework. However, it would be more systematic if these results were given within the research hypotheses/questions.

5. Table 1 presents information on the proportion of female participants as well as countries. However, while a detailed discussion addressing the female proportion was made in the discussion section, the same was not done for countries. In this case, I can not sure what the purpose of collecting and publishing data on countries was.

6. I believe the limitations of the manuscript should also have been expressed by the researchers.

6. PLOS authors have the option to publish the peer review history of their article (what does this mean?). If published, this will include your full peer review and any attached files.

Reviewer #1: **Yes: **Dr Mohit Kumar Patralekh

Reviewer #2: **Yes: **Eyüp Yılmaz

---

## [Author Response · Author response to Decision Letter 0]

29 Dec 2023

RESPONSE TO THE EDITOR

Dear Dr. Charland,

Thank you for your review. We have made the necessary revisions to address the comments and corrections raised during the review process.

Attached are the requested documents:

Rebuttal letter labeled 'Response to Reviewers'

Marked-up copy of the manuscript labeled 'Revised Manuscript with Track Changes'

Unmarked version of the revised manuscript labeled 'Manuscript'

We are open to further revisions if needed and appreciate your guidance. The updated financial disclosure is included in the cover letter.

Thank you again for your remarks.

Best regards,

Patrycja Kiszka on behalf of the authors

RESPONSE TO REVIEWER 1 

16 December 2023 

Dear Dr Mohit Kumar Patralekh, 

I am writing this comment in response to the review and suggestions that I received on Novermber 2nd, 2023. All of the comments were used to make the manuscript more communicative and meaningful. We would like to thank you for your diligent and constructive review. 

Please find the revised version of the manuscript The tangled ways to classify games: A systematic review of how games are classified in psychological research. I include the manuscript in marked version. 

In the following section you can find detailed comments for your suggestions. 

1. Was the protocol for this systematic review preregistered( on any site like PROSPERO) or pre-published? Please provide details.

According to PLOS One “Criteria for publication” and PRISMA guidelines, there is no information about obligation of registration. To the best of our knowledge, the preregistrations are recommended more for the research studies than to the reviews. Consequently, our systematic review was not preregistered. The information was added to the manuscript.

2. Detailed search strategy for at lease one database may be provided.

The section of search strategy has been transformed to present detailed information about the selected databases, the filters applied regarding the subject, keywords and the selection process. Also described are the exclusion criteria, which previously only appeared on the PRISMA diagram.

3. The methods section says "The first reviewer screened ScienceDirect and Taylor and Francis records, whereas the second reviewer focused on Scopus and PubMed records." Usually all databases are searched by two reviewers and articles are also selected by both of them, and any discrepancy is resolved by third reviewer. The manuscript is actually silent on whether screening and selection was done by two reviewers.

Thank you for a very significant comment. In order to eliminate potential bias of independent decision of selection, we conducted a cross-screening which meant re-screening the database results by another reviewer. As a consequence, the databases were screened twice, systematically and independently by two reviewers. In case of discrepancies, they were resolved through clarification and consensus. This information has been added to the method description in the search strategy section.

We would like to inform you that we have updated the literature review for an additional year, extending the coverage from 2012-2022 to 2012-2023. This enhancement ensures that the review is as current as possible and will prove valuable to its audience.

If further improvement is needed, I am at your disposal. 

With the best regards, 

Patrycja Kiszka on behalf of the authors 

 

RESPONSE TO REVIEWER 2 

16 December 2023 

Dear Prof Eyüp Yılmaz, 

I am writing this comment in response to the review and suggestions that I received on Novermber 2nd, 2023. All of the comments were used to make the manuscript more communicative and meaningful. We would like to thank you for your diligent and constructive review, and we also appreciate pointing out specific pieces of literature that we should have included.

Please find the revised version of the manuscript The tangled ways to classify games: A systematic review of how games are classified in psychological research. I include the manuscript in marked version. 

In the following section you can find detailed comments for your suggestions. 

1. I wonder why the researchers did not include the terms "digital game" and the adjacent "videogame" in their search strategies. Because although the term "video game" is predominantly preferred, we also encounter the use of "digital game" and "videogame" in the literature. This not only leads to the exclusion of studies that meet the criteria but also reduces the content validity of the research. For example, Is there any possibility that the study below was not encountered for this reason and could not be included in the manuscript? Because it seems, the videogames were categorized by their genres as well as traditional games in the study. "Yılmaz, E., Yel, S., & Griffiths, M. D. (2022). Comparison of children's social problem-solving skills who play videogames and traditional games: A cross-cultural study. Computers & Education, 187, 104548.

In fact, the study (Yilmaz el al., 2022) was not included in the review because of the keyword “video games” with spacebar. According to your comment, for this reason we may have missed relevant studies. Therefore, we decided to conduct a complementary search with the following keywords: “video game genre” OR “video game classification” OR “video game categories” OR “videogame genre” OR “videogame classification” OR “videogame categories” OR “digital game genre” OR “digital game classification” OR “digital game categories”. Additionally, we agreed to expand the temporal frame of the search to include publications from 2023. By doing so, we maximize the chances that all relevant and up-to-date articles are included.

2. It appears that the databases were searched separately by two different researchers and that a single researcher decided on the studies to be included. There is a problem here in terms of coder reliability. Each database should have been screened by at least two researchers in case of an oversight or indecision, and the procedure followed in case of disagreements should have been explained.

This note is similar with reviewer 1 and we found those very relevant. In order to eliminate potential bias of independent decision of selection, we conducted a cross-screening which meant re-screening the database results by another reviewer. As a consequence, the databases were screened twice, systematically and independently by two reviewers. In case of discrepancies, they were resolved through clarification and consensus. This information has been added to the method description in the search strategy section.

3. The researchers need to add not only years but also months to the period (i.e. January 2012 to December 2022 (inclusive)).

We agree. We have completed the detailed information regarding the time range considered for the search. We would like to inform you that we have updated the literature review for an additional year, extending the coverage from 2012-2022 to 2012-2023. This enhancement ensures that the review is as current as possible and will prove valuable to its audience. This is stated in the first point of the inclusion criterion: “The articles were published between January 2012 and December 2023 (inclusively).”

4. The results are given within a certain logical framework. However, it would be more systematic if these results were given within the research hypotheses/questions.

We have taken this suggestion into account through amendments. According to the aim of the study which includes capturing the most recent and the most prominent methods of classifying video game genres in empirical studies, we decided to move the section “Quality of creating game taxonomies” to the top of the results. Then, we provided the parts: ”General characteristics of taxonomies”, “One genre, multiple definitions”, “The multi-genre misunderstanding”, “Steam tags – different approach to game classification”.

5. Table 1 presents information on the proportion of female participants as well as countries. However, while a detailed discussion addressing the female proportion was made in the discussion section, the same was not done for countries. In this case, I can not sure what the purpose of collecting and publishing data on countries was.

Indeed, with the large amount of information, we overlooked the discussion of the geographic coverage of the included studies. This section has been supplemented in the manuscript.

6. I believe the limitations of the manuscript should also have been expressed by the researchers.

Thank you for the suggestion, we agree. We have added the "Limitations" section to our manuscript.

If further improvement is needed, I am at your disposal. 

With the best regards, 

Patrycja Kiszka on behalf of the authors

---

## [Decision Letter · Decision Letter 1]

5 Feb 2024

PONE-D-23-27687R1The tangled ways to classify games: A systematic review of how games are classified in psychological researchPLOS ONE

Dear Dr. Kiszka,

Thank you for submitting your manuscript to PLOS ONE. After careful consideration, we feel that it has merit but does not fully meet PLOS ONE’s publication criteria as it currently stands. Therefore, we invite you to submit a revised version of the manuscript that addresses the points raised during the review process.

We look forward to receiving your revised manuscript.

Kind regards,

Patrick Charland

Academic Editor

PLOS ONE

Journal Requirements:

Additional Editor Comments:

I am pleased to observe that the majority of comments have been incorporated, and the reviewers have generally accepted most of the elements identified in the initial submission. There remain one or two minor points to clarify. However, I would like to draw your attention to the fact that the format of your references needs to be revised to conform to the Vancouver style. Please see the references section in the submission guidelines at: https://journals.plos.org/plosone/s/submission-guidelines

Reviewers' comments:

Reviewer's Responses to Questions

**Comments to the Author**

1. If the authors have adequately addressed your comments raised in a previous round of review and you feel that this manuscript is now acceptable for publication, you may indicate that here to bypass the “Comments to the Author” section, enter your conflict of interest statement in the “Confidential to Editor” section, and submit your "Accept" recommendation.

Reviewer #1: All comments have been addressed

Reviewer #2: All comments have been addressed

2. Is the manuscript technically sound, and do the data support the conclusions?

Reviewer #1: Yes

Reviewer #2: Yes

3. Has the statistical analysis been performed appropriately and rigorously? 

Reviewer #1: Yes

Reviewer #2: N/A

4. Have the authors made all data underlying the findings in their manuscript fully available?

Reviewer #1: Yes

Reviewer #2: Yes

5. Is the manuscript presented in an intelligible fashion and written in standard English?

Reviewer #1: Yes

Reviewer #2: Yes

6. Review Comments to the Author

Reviewer #1: Thanks for the response and for revising the manuscript as per suggestions. Please check why it's suitable to mention the search period as 1full year (January 2023 to December 2023). Rather dates when the last search was performed may be mentioned.

Reviewer #2: Dear Authors,

Thank you for your meticulous handling of all correction. I think your manuscript is now publishable after a minor revision. It seem that in two papers (Yılmaz et al., 2022 a-b) "constant" variable which was obtained as a result of multiple regression analysis was considered as a videogame genre. The concept of "constant" should be removed from the game genres. Otherwise, it would be a big mistake to publish the manuscript as it is.

All the best.

7. PLOS authors have the option to publish the peer review history of their article (what does this mean?). If published, this will include your full peer review and any attached files.

Reviewer #1: **Yes: **Dr Mohit Kumar Patralekh

Reviewer #2: **Yes: **Eyüp Yılmaz

---

## [Author Response · Author response to Decision Letter 1]

14 Feb 2024

RESPONSE TO REVIEWER 1 

11 February 2024 

Dear Dr Mohit Kumar Patralekh, 

I am writing this comment in response to the review and suggestions that I received on February 5th, 2024. We appreciate your feedback and suggestion regarding the mention of the search period in the manuscript. Upon reconsideration, we agree that it would be more appropriate to specify the dates when the last search was performed rather than stating a full year. We will make the necessary adjustments to accurately reflect the timing of the search process in the revised manuscript.

Please find the revised version of the manuscript The tangled ways to classify games: A systematic review of how games are classified in psychological research. I include the manuscript in marked version. 

If further improvement is needed, I am at your disposal. 

With the best regards, 

Patrycja Kiszka on behalf of the authors 

 

RESPONSE TO REVIEWER 2 

11 February 2024 

Dear Prof Eyüp Yılmaz, 

I am writing this comment in response to the review and suggestions that I received on February 5th, 2024. Thank you for your thorough review and acknowledgment of our efforts in addressing the corrections. Indeed, we concur with your observation. The oversight regarding the inclusion of the "constant" variable has been rectified, and both the manuscript and the supplementary materials have been amended accordingly. We appreciate your keen attention to detail, which has undoubtedly enhanced the quality and integrity of our work.

Please find the revised version of the manuscript The tangled ways to classify games: A systematic review of how games are classified in psychological research. I include the manuscript in marked version. 

If further improvement is needed, I am at your disposal. 

With the best regards, 

Patrycja Kiszka on behalf of the authors

---

## [Editor Report · Decision Letter 2]

16 Feb 2024

The tangled ways to classify games: A systematic review of how games are classified in psychological research

PONE-D-23-27687R2

Dear Dr. Kiszka,

We’re pleased to inform you that your manuscript has been judged scientifically suitable for publication and will be formally accepted for publication once it meets all outstanding technical requirements.

Kind regards,

Patrick Charland

Academic Editor

PLOS ONE
---

## [Editor Report · Acceptance letter]

13 Jun 2024

PONE-D-23-27687R2 

PLOS ONE

Dear Dr. Kiszka, 

I'm pleased to inform you that your manuscript has been deemed suitable for publication in PLOS ONE. Congratulations! Your manuscript is now being handed over to our production team.

Kind regards, 

on behalf of

Dr. Patrick Charland 

Academic Editor

PLOS ONE